

# Heart rate variability over the decades: a scoping review

Amina Sundas[1], Ivan Contreras[1,2], Judith Navarro-Otano[3], Júlia Soler[1], Aleix Beneyto[1] and Josep Vehi[1,4]

[1] Modeling Intelligent Control Engineering Laboratory, Institut d'Informatica i Aplicacions, Universitat de Girona, Spain, Girona, Spain
[2] Universitat de Girona, Spain, Girona, Spain
[3] Neurology Service, Hospital Clinic, Barcelona, Spain, Barcelona, Spain
[4] Centro de Investigación Biomédica en Red de Diabetes y Enfermedades Metabólicas Asociadas (CIBERDEM), Girona, Spain

## ABSTRACT

Heart rate variability (HRV) measures the variation in the time interval between successive heartbeats, reflecting the influence of the autonomic nervous system (ANS) on heart rate (HR) changes. This review provides an extensive overview of HRV measurement techniques, their applications, and their limitations in healthcare, exploring their potential for prognosis and condition assessment. A scoping review was conducted, gathering literature reviews on HRV spanning the past fifty years using PubMed, Scopus, and Web of Science databases. Our findings identified significant research gaps, including contradictions in the literature and the absence of standardized protocols for HRV measurements, which complicate the establishment of consistent baseline values. Additionally, the lack of protocols for pulse rate variability (PRV) in the context of advanced sensor technology hinders progress in HRV research. Despite these challenges, HRV remains significant in assessing cardiac autonomic function and its potential clinical applications. However, barriers such as device unfamiliarity, data accuracy concerns, and a lack of clinical trials limit its adoption. Further research is required to elucidate the relationship between abnormal HRV and health problems and to establish consistent baseline values for advancing HRV applications.

## INTRODUCTION

Heart rate variability (HRV) refers to the variation in the time interval between successive heartbeats. It is a measure of the naturally occurring fluctuations in the timing of each heartbeat and is controlled by the autonomic nervous system (ANS). HRV is a measure that helps us understand how our heart rate (HR) changes over time. If you calculate the average time between consecutive heartbeats, you obtain the resting beat-to-beat interval. HRV describes the variations in this resting beat-to-beat interval. There may be variations in the range of 10% to 30% in the beat-to-beat timing interval even when the HR per

Corresponding author
Ivan Contreras,
ivan.contreras@udg.edu

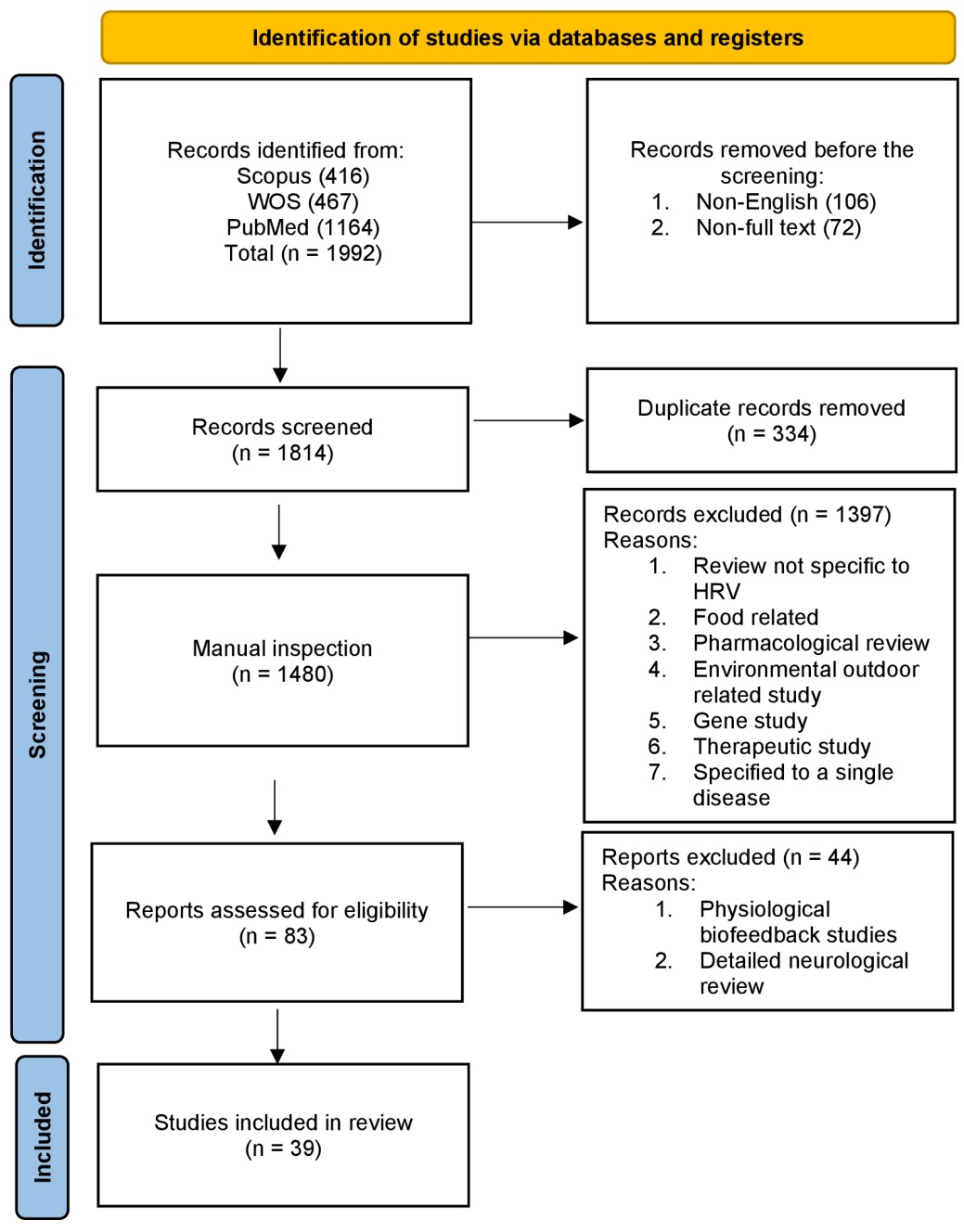

**Figure 1** Overview of the scoping review process using PRISMA 2020 flow diagram guidelines.

minute does not change. We conducted a scoping review (see Fig. 1) of the field to identify the measurement methods used in past and current research into HRV.

Our understanding begins with the autonomic nervous system (ANS), which is mainly composed of two branches: the parasympathetic nervous system (PNS) and the sympathetic nervous system (SNS). The SNS, also called the fight-or-flight system, activates in stressful situations leading to increasing HR, constricting blood vessels, and elevating blood pressure

among others. Conversely, the PNS, known as the rest-and-digest mechanism, opposes the SNS by decreasing the HR and lowering blood pressure. Both the SNS and PNS work in harmony to maintain a sympathovagal balance, ensuring overall well-being (*Gabella, 2012*). An imbalance between these systems can signify heart-related irregularities (*Ishaque, Khan & Krishnan, 2021*). The heart is innervated by both the SNS and PNS at the sinoatrial (SA) node, often referred to as the heart's pacemaker. This node orchestrates the rhythm of our heartbeats, subject to the influence of the PNS and SNS. The PNS can lower HR by as much as 50 beats per minute, whereas the SNS can elevate HR to 200 beats per minute. In the absence of PNS and SNS influence, the heart maintains an intrinsic HR of 100-110 beats per minute (*Alugubelli, Abuissa & Roka, 2022*). Furthermore, our resting HR typically hovers between 60 and 70 beats per minute, signifying a predominance of PNS mediation at rest. The two limbs of the ANS constantly interact to keep the body's cardiac output at an adequate level to suit its demands. Therefore, measuring HRV can indirectly aid in understanding autonomic function (*Tiwari et al., 2021*; *Peltola, 2012*).

The idea that a healthy HR must have variations was established a century ago (*Draghici & Taylor, 2016*). As HR is largely mediated by both the PNS and SNS, HRV is considered a more sensitive index of the ANS function than simple HR analysis (*Routledge, Chowdhary & Townend, 2002*). Resting HR is predominantly regulated by the PNS through the vagus nerve, which modulates HR in synchrony with the respiratory cycle, as evidenced by the physiological respiratory sinus arrhythmia (RSA). The SA node receives input from both the PNS and the SNS, with the vagal effects occurring more rapidly than the slower-acting sympathetic effects. HRV reflects primarily vagal modulation of the heart (*Routledge, Chowdhary & Townend, 2002*; *Draghici & Taylor, 2016*).

The complexity of heart function is highlighted by the beat-to-beat variability in cardiovascular signals such as HR, arterial blood pressure, and stroke volume. These fluctuations arise from a dynamic interplay of physiological processes, influenced by both external factors and internal regulatory mechanisms. For example, *Kreibig (2010)* emphasized that emotional stimuli, such as joy and anxiety, do not produce uniform HR or HRV responses, as these are shaped by the complex interplay between emotional context, individual interpretation, and physiological processes. The ANS plays a critical role in modulating these processes, particularly through the baroreceptor reflex. Additionally, variations in ECG morphology reflect mechanical influences and can indicate cardiac stability or arrhythmia susceptibility. Understanding these dynamics is essential for assessing cardiovascular health, as diminished variability is linked to increased mortality risk (*Appel et al., 1989*).

HRV can be assessed through two primary methods: electrocardiogram (ECG) or photoplethysmogram (PPG) sensors. Figure 2 illustrates the time between consecutive heartbeats using both ECG and PPG signals, providing a visual representation of these measurement techniques. ECG is a widely employed non-invasive diagnostic technique due to its ease of measuring electrical activity and strong correlation with the heart's mechanical and metabolic activity. Most clinical and commercial ECG-based HR monitors use algorithms to recognize the QRS complex on an electrocardiogram (see Fig. 2). On the other hand, PPG relies on measuring changes in microvascular blood volumes (*Alugubelli,*

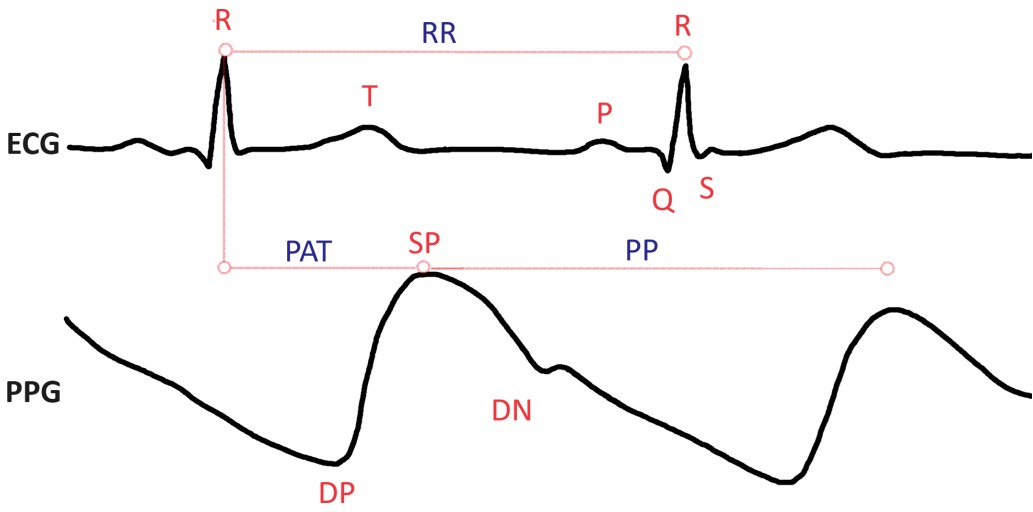

**Figure 2** **Heart rate variability, as presented in electrocardiogram (ECG) and photoplethysmogram (PPG) signals. The ECG signal is composed of the P, Q, R, S, and T waves; where P shows atrial depolarization, Q, R, and S are termed together as the QRS complex and show ventricular depolarization of the heart and the T wave shows ventricular repolarization.** The distance between R peaks is measured in milliseconds as the RR interval. Similarly in the PPG signal, the SP (systolic peak) represents the maximum point of blood volume increase following a heartbeat. The DN (dicrotic notch) is a small dip following the SP, which corresponds to the closure of the aortic valve. The DP (diastolic peak) follows the dicrotic notch and indicates the secondary increase in blood volume. The PP (pulse interval) is the time between successive systolic peaks, measured in milliseconds. PAT (pulse arrival time) is the interval between the R-peak of the ECG signal and the SP of the corresponding PPG signal.

*Abuissa & Roka, 2022*; *Hinde, White & Armstrong, 2021*). When HRV is determined using the PPG method, it is often called pulse rate variability (PRV) (*Yuda et al., 2020*). PRV is commonly integrated into smartwatches, fitness bands, and various portable devices, making it more accessible for continuous monitoring and health tracking. Data from both ECG and PPG methods can be transformed into a tachogram recording for further analysis. Indeed since the introduction of the first pulse measuring device by Anastasios Filadelfeus in the nineteenth century, besides ECG as a measurement method, there are other methodological approaches in HR and HRV research, such as measurement with a chest strap or light-based technology for measuring blood volume pulse. Although ECG is still the most common method used to measure HRV, PPG sensors are becoming increasingly common. Innovative methods such as smart watches or clothing, chest straps, and ear-canal sensors can be used to measure HRV with PPG's assistance. The advantage is increased data collection convenience and accuracy (*Stangl & Riedl, 2022*).

The RR intervals of the QRS complex in the ECG are modulated at different frequencies by the differential rhythmic contributions from sympathetic and parasympathetic autonomic activity. This modulation forms the basis for HRV analysis, which can be performed using three main techniques: time-domain, frequency-domain, and nonlinear methods, each offering unique insights into cardiac rhythms. One of the most significant advantages of HRV analysis is its ability to distinguish between sympathetic and parasympathetic influences on HR since it has significant implications for understanding

ANS function and its impact on cardiovascular health. Power spectrum analysis of HRV offers a sensitive, quantitative, and noninvasive method for assessing short-term cardiovascular control (*Akselrod et al., 1981*). This differentiation can provide valuable information in various clinical and research contexts, from assessing stress levels to evaluating the risk of cardiac events. As technology continues to advance, the integration of these sophisticated HRV analysis techniques into wearable devices and healthcare systems is expanding our capacity to monitor and interpret the activity of the ANS in real time and in everyday settings. This progress holds promise for improving early detection of cardiovascular problems, personalizing treatment approaches, and enhancing our overall understanding of heart-brain interactions and their impact on health and well-being.

This review article presents an analysis of the literature published in the field of HRV and its applications in the past 50 years, guided by the research question: What are the primary measurement methods for HRV, how have they evolved over the past 50 years, and what are the key challenges and opportunities in their application for clinical and technological advancements? This paper is for researchers and healthcare professionals interested in HRV and its role in health. It also speaks to those looking into improving HRV measurement methods and wearable technology for practical use. Moreover, it is relevant for professionals developing standardized protocols and improving measurement accuracy in HRV research and technology. We have addressed the relevant aspects of the area, including the identification of the main research gaps—such as the lack of standardized measurement protocols, challenges in data interpretation, and the need for further validation of wearable HRV measurement devices—as well as the search for appropriate outcome metrics, and current methodological techniques, all aimed at presenting a comprehensive overview of state-of-the-art HRV measurement techniques. This includes their applications across various fields of healthcare and the challenges faced by engineers in utilizing HRV values, alongside discussions on the potential of HRV data for patient prognosis and condition assessment, as well as the opportunities presented by wearable HRV and PRV measuring devices.

## SURVEY METHODOLOGY

Given the extensive body of literature on HRV, we crafted a comprehensive scoping review of reviews to provide a holistic overview of HRV measurement techniques, their varied applications, and the associated limitations in utilizing HRV values. The reason for choosing a scoping review was to synthesize a broad and diverse body of literature, map key concepts, and identify research gaps in HRV research. Unlike systematic reviews, this method is especially well-suited for offering an exploratory overview of complex topics without concentrating only on predetermined outcomes. Scoping reviews are not only about summarizing existing literature but also about critically assessing it to uncover areas that require more research and help researchers understand the landscape of a particular field (*Schryen et al., 2020*). To assemble our dataset, we conducted an initial screening process using three core databases: PubMed, Web of Science (WOS), and Scopus. Our search criteria were precise, focusing on the English language, human studies, and full-text

reviews with no time limit on the search and exclusively related to HRV. These reviews had to specifically address the clinical relevance of HRV, its history and its integration with wearable technology. The study selection process, summarized in Fig. 1, was a critical phase that involved scrutinizing and refining the initial pool of articles based on strict criteria that cover the following:

- Initial count: The initial search yielded a total of 1,992 reviews that met the basic search criteria. Search terms included: ("heart rate variability" OR "HRV").
- Duplicate removal: To streamline the dataset, we utilized Bibliometrix to identify and remove 334 duplicate entries.
- Exclusion criteria: Articles were excluded from consideration if they fell into the following categories: therapeutic studies, gene/genetic studies, detailed neurological reviews, reviews exclusive to a specific disease, psychological biofeedback studies, as well as those focused on food, medicine, and environmental pollution. The exclusion of these articles was done to narrow the scope of the review and keep the focus on HRV measurement techniques, clinical relevance, and applications, ensuring that the findings are broadly applicable.
- Second screening: 83 articles were downloaded, which were free from duplicates, and subjected to a more in-depth evaluation through a full-article reading.
- Final selection: After the second screening, 39 articles emerged as the select few that fulfilled our rigorous criteria and were included in our scoping review, enabling us to provide an indepth overview of HRV research developments over the years. We also included device information from their respective websites, which might be updated over time. In addition, the scoping review method allowed us to explore HRV literature comprehensively and identify trends and gaps across a broad spectrum of research. This methodology prioritizes breadth and exploratory analysis, making it ideal for addressing the study objectives (*Schryen et al., 2020*). Figure 1 shows the PRISMA methodology flow diagram.

## RESULTS

The results section consists of seven main subsections. The first subsection captures the history of HRV in the literature and explains the evolution of HRV measuring devices in the last fifty years. The second and third subsections find existing methodological approaches to measuring and analyzing HRV, which help in understanding HRV terminologies used in the literature. We then started to identify research gaps and determine appropriate and consistent HRV terminologies. The fourth one explores the confounding factors that affect the value of HRV, the fifth subsection explains the clinical uses of HRV values, the sixth subsection explains some of the contradictions in the literature about HRV parameters and the seventh subsection elaborates on the potential of using HRV parameters.

### HRV through the decades
The history of HRV captured in the articles reviewed reveals a fascinating evolution in our understanding of this physiological phenomenon. In 1733, Stephen Hales made

a groundbreaking discovery by identifying HRV in animals. Hales' experiments on arterial blood pressure variations during respiration introduced systematic measurement techniques. Later, Carl Ludwig used a kymograph to study the frequency and pulse waves in dogs and reported a connection between variations in heartbeat and breathing (RSA) (*Ernst, 2017*). Ludwig's use of the kymograph advanced the understanding of heartbeat fluctuations. Despite significant challenges in early HRV studies, such as manual RR interval measurements, inadequate sampling rates, and lack of standardized protocols, their work laid the groundwork for modern advancements. Ludwig's is also credited with describing the link between heartbeat fluctuations and respiration (RSA). This approach was innovative for its time and contributed to the foundational understanding of cardiovascular physiology (*Ernst, 2017*). Advancements continued with the development and widespread adoption of electrocardiograms (ECGs) by Willem Einthoven in the early 1900s, allowing for precise heart rhythm recording. This was followed by the introduction of Holter monitors in the 1950s, providing the capability for continuous monitoring of cardiac activities. In 1961, scientists initiated time-domain analysis of beat-to-beat variations, setting the stage for more comprehensive investigations (*Billman, 2011*).

It was in 1965 when Hon and Lee made a pivotal discovery by linking reduced HRV around delivery time to increased fetal distress, highlighting the clinical relevance of HRV (*De Maria et al., 2021*; *Stys & Stys, 1998*). They identified alterations in the beat-to-beat interval as the earliest indicators of fetal distress, preceding changes in HR itself. The term 'heart rate variability' was later popularized by their findings and subsequent research (*Stys & Stys, 1998*). Early studies from 1967 revived clinical research interests in obstetrics, gynaecology, and cardiology, emphasizing its importance in assessing neurological function and brain-vagal-heart communication (*Stein et al., 1994*).

As far back as the 1970s, physiologists began uncovering crucial factors linked to HRV. They discovered that several factors interact with one another to influence HRV. Notably, they identified key factors such as the control centres in the brainstem that regulate the ANS, the baroreceptors responsible for stabilizing blood pressure, and the impact of breathing patterns. Since the early 1970s, changes in HRV were accepted as a marker of disease when the initial research connecting HRV to diabetic autonomic neuropathy was published (*Kautzner & John Camm, 1997*). Eight disparate time-domain parameters were introduced by Luczak and Laurig and 26 indices for the time-domain were proposed by Opmeer (*Kobayashi, Ishibashi & Noguchi, 1999*). Significant changes occurred in the late 1970s when researchers began focusing on RR intervals for a more in-depth analysis of HRV using ECG. This led to the creation of time-domain metrics like the standard deviation of all NN intervals (SDNN) or the root mean squared value of the successive differences of the interbeat interval(RMSSD) (*Ishaque, Khan & Krishnan, 2021*). A detailed explanation of these terms can be found in 'HRV analysis techniques'.

In 1981, a study found that analyzing RR interval variability provides physiological insights into cardiac autonomic control (*Ishaque, Khan & Krishnan, 2021*). It was observed that HRV disappeared after high-dose atropine blockade, confirming its relationship with autonomic control. By the 1990s, RR intervals were considered less efficient than spectral analysis methods, leading to an increase in studies centred on power spectral

density metrics such as low frequency (LF), high frequency (HF), and the LF/HF ratio related to ANS impairment caused by heart problems (*Ishaque, Khan & Krishnan, 2021*). In the 1990s, Holter systems had already modified their equipment with options for the assessment of HRV (*Malik & Camm, 1990*). The 1996 consensus report by the Task Force of the North American Society of Pacing and Electrocardiology and the European Society of Cardiology (NASPE/ESC) established standards for measuring, interpreting, and usage of HRV measurements for clinical applications (*Task Force of the European Society of Cardiology the North American Society of Pacing Electrophysiology, 1996*).

Since the early 2000s, HRV has become a promising physiological indicator for tracking athletic performance and adaptations to exercise (*Dobbs et al., 2019*). HRV, traditionally determined through medical ECG recordings and specialized software, has been limited to laboratory or clinical settings due to high costs and technical knowledge. However, advances in technology and portable monitoring devices have led to many commercial systems now including HRV as a feature, which was validated as early as 2003. In 2006, Poincaré plots were introduced to visually represent non-linear scatter plots linked to heart conditions and reduced HRV (*Ishaque, Khan & Krishnan, 2021*). Today, in the 2020s, HRV knowledge is considered essential by athletes and technology entrepreneurs for monitoring body recovery, optimizing exercise routines, gauging sleep quality, and managing stress.

The history of HRV is reflected in Fig. 3, which shows the trends in scientific publications on this topic over the last five decades. This figure provides valuable insight into the evolution of perspectives on HRV and shows how it has become a fundamental tool for understanding and improving human health and performance. In the 1970s and 1980s, a significant percentage of HRV studies were focused on gynecology, particularly on fetal HR monitoring, which became a standard of care and contributed to reduced morbidity. This period marked the emergence of HRV's first clinical applications, with its utility in monitoring fetal and maternal health being widely recognized. By the 1980s, evidence of complex non-linear dynamics in HRV further expanded the understanding of HRV's physiological significance.

With a growing interest in cardiovascular health in the 1990s, HRV research began to shift toward physiology and sports sciences, emphasizing its role in autonomic regulation and cardiovascular conditions. The standardization of HRV measurement and calculation methods in 1996 played a crucial role in facilitating this expansion, making investigations more comparable and enabling meta-analyses across various studies (*Task Force of the European Society of Cardiology the North American Society of Pacing Electrophysiology, 1996*). HRV research shifted from a focus on gynecology, particularly in fetal HRV monitoring, to cardiovascular health and physiology in the 1990s was driven by advancements in ECG technology and growing recognition of HRV's role in autonomic regulation (*Billman, 2011*). This period also saw an exponential growth in HRV research, with over 14,000 articles mentioning HRV and more than 2,000 clinical trials exploring its applications (*Ernst, 2017*).

By the late 2000s, HRV research had diversified further, with most studies focusing on sports sciences, biomedical engineering, and physiology. With the innovations like introducing HR monitoring and oxygen saturation through watches in mid 2000s

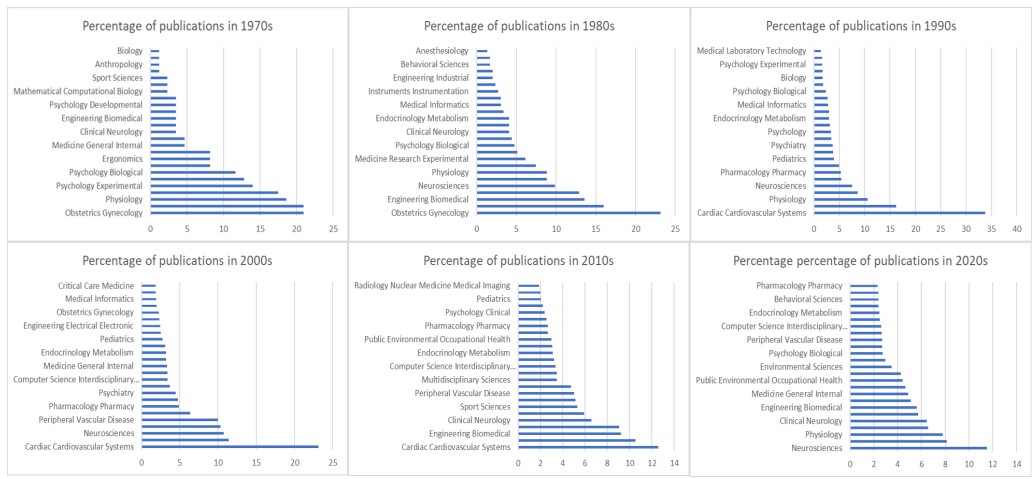

**Figure 3** Evolution of HRV research over five decades, reflecting its growing interdisciplinary importance.

led to explosion in the consumer wearables (*Alugubelli, Abuissa & Roka, 2022*). The interdisciplinary nature of HRV research became more evident as it was increasingly applied to areas such as optimizing athletic performance, understanding autonomic regulation, and advancing wearable technologies. This progression underscores the continued growth and broadening scope of HRV research, which has become a critical tool in understanding and improving human health and performance.

## HRV measurement and devices

Progress in the adoption and analysis of HRV has been coupled with corresponding progress in HRV measuring device technology. Historically, auscultation has been used by doctors for many hundred years to identify cardiac tones and heartbeat rhythms, and medical professionals have long been aware of changes in the beat-to-beat rhythm brought on by age, sickness, and psychological conditions. The scientific study of beat-to-beat HR rhythms was postponed until certain technological developments made it possible to precisely and consistently measure the electrical activity of the heart. The invention of four devices led to the advancement of this field: galvanometers, kymographs, ink-writing polygraphs, and ECGs. Therefore, the age of scientific examination of HRV was dependent on the availability of the electrocardiograph for physiological and clinical research. Accurate timing of beat-to-beat variability was dependent upon the detection of features of the ECG (*i.e.,* the peak of the R-wave). Engineers used electrical circuits and clocks that later became tachometers to locate the peak of R-waves. Timing improved with the introduction of laboratory computers, and computer algorithms were able to recognize R-waves and other ECG components (*Ernst, 2017*; *Billman, 2011*).

The early iterations of ambulatory cardiac monitors were relatively basic. These devices were primarily focused on capturing and transmitting data related to a patient's cardiac activity without much onboard processing. A significant advancement came
with the introduction of Holter monitors, which allowed for local data storage. Holter monitors utilized cassette recorders to store the recorded cardiac data, enabling healthcare professionals to review and analyze the patient's heart rhythms at a later time. This was a crucial development as it offered a way to capture long-term data, which was essential for diagnosing irregular heart rhythms and other cardiac conditions. The heart rhythms were evaluated using ECG recordings from a Holter monitor to offer a sensitive, noninvasive evaluation of autonomic input to the heart (*van Ravenswaaij-Arts et al., 1993*; *Stein et al., 1994*). By assessing HRV on 24-hour Holter recordings, autonomic function during the day and at night was evaluated (*Kristal-Boneh et al., 1995*). The analysis was typically based on either long-term Holter ECG recordings or short-term ECG recordings (*Cygankiewicz & Zareba, 2013*).

In the 1960s, portable Holter devices were introduced, enabling semiautomatic electrocardiogram analysis. The 1970s marked the beginning of computer-based automated pattern recognition for ECG analysis. Until the 1980s, cardiotachometers were commonly employed to capture a person's electrical signal and record their HR for HRV studies. During the 1980s, 2-lead ECGs were typically utilized in HRV research. Starting in the 1990s, HRV research expanded to include diverse methods. PPG and thoracic belts were utilized to analyze HRV by measuring blood pressure and respiration. The 1990s brought about increased storage capacity with digital storage and the standardization of data formats. The introduction of twelve-lead ECGs in 2000 enabled more effective analysis of various heart conditions by researchers collaborating with clinicians (*Ishaque, Khan & Krishnan, 2021*). The next major leap in technology was the miniaturization of computers. Ambulatory cardiac monitors have evolved from simple wireless transmitters to sophisticated devices capable of local data storage, signal processing, and analysis. The integration of telecommunication services has further revolutionized the field, enabling real-time data transmission and advanced computational analysis, ultimately enhancing the monitoring and diagnosis of cardiac conditions (*Alugubelli, Abuissa & Roka, 2022*). By the 2010s, consumer-grade remote monitoring became accessible to the general public, and in the 2020s, technology is advancing towards cloud-based solutions and the integration of machine learning and artificial intelligence for signal analysis and interpretation.

Nowadays, HRV is predominantly measured using two techniques: the ECG-based technique and the PPG-based techniques both of which utilize various types of HR monitors. Since the 1920s, surface electrodes have been used in clinical practice to directly measure cardiac electrical activity. The electrode-skin interface greatly influences the quality of the signal. Medical-grade wearable ECG-based sensors typically record data from multiple channels and enable more in-depth analyses, such as assessing conduction problems, the location of premature beats, and ischemia. Initially, HRV was quantified manually using computations of the mean RR interval and its short-term standard deviation, or SDNN. However, in the 1990s, improvements in recording methods, computing power accessibility, and inventive microprocessor technology made it possible to analyze long-term records. ECG sensors are still considered the best technology for measuring HR and HRV.

PPG was developed in the 1930s. In a tachogram recording of PPG, the duration from one peak of the blood volume curve to the next peak on the PPG signal represents the beat-to-beat interval (*Faust et al., 2022*). PPG has been reintroduced due to advancements in semiconductor technology, optoelectronics, and digital signal processing. Current PPG sensors use low-cost optoelectronic components in red and/or near-infrared wavelengths. PPG technology is versatile and useful in various clinical applications. It offers advantages over ECG-based HRV analysis, especially in clinical situations where a pulse oximeter is already present. This allows HRV analysis to be included without requiring an ECG, providing a significant advantage (*Schäfer & Vagedes, 2013*). Consumer-grade devices that use PPG have a good association with ECG readings. However, PPG-based medical-grade devices are more accurate (*Alugubelli, Abuissa & Roka, 2022*; *Dobbs et al., 2019*; *Hinde, White & Armstrong, 2021*).

As discussed above, the majority of commercial monitoring devices employ either PPG or ECG sensors. The use of PPG-based monitors has increased in wearable devices such as smartwatches, fitness bands, and other portable devices (*Alugubelli, Abuissa & Roka, 2022*; *Hinde, White & Armstrong, 2021*). The data collected by these devices is then analyzed using various techniques. The accessibility of high-quality sensors and the slow data rate of HR signals contribute to the ease of measuring, communicating, storing, and processing HRV data. However, compared to HRV values obtained from an ECG machine, there has been a percentage of error associated with HRV measurements obtained from portable devices. Not all portable devices, whether PPG-based or single lead ECG, report a similar error measurement to that of an ECG machine but the literature also showed that this error is independent of the type of device and is primarily affected by the metric, position, and biological sex. Some metrics obtained through spectral analysis have shown poor agreement with the gold standard ECG machine metrics specifically the LF metric. Although time-domain measurements obtained through portable devices, like RMSSD, have been found to have good agreement with the gold standard ECG measurements. The use of these portable devices seems to be a sensible choice in terms of the usability of HRV measurements since it lowers the cost of the measurement and makes it simpler and easier to use (*Dobbs et al., 2019*).

Regarding sampling, some HRV measurement devices take spot measurements, whereas others only take measurements during periods of rest and/or sleep. Few devices are capable of measuring HRV continuously for 24 h. Wearable technology includes armbands, PPG wristwatches, and single-channel ECG chest wraps that measure PRV and HRV, respectively. Although some of these devices' accuracy is validated against the gold standard ECG values, others still require additional validation.

A few of the devices that have been examined are included in Table S1. The table also specifies which devices have been approved by the FDA and have been tested against the gold standard ECG measurements. Table S1 contains information related to contemporary wearable devices with HRV measuring technology. The information on the manufacturing year, if the device is tested against the gold standard and FDA approval is taken from their respective sites. Although wearable HRV measuring technology has become popular, there is only a small amount of independent clinical data available on it. However, not all of

these wearable devices are officially approved for use by healthcare professionals. In 2021, out of 45 different devices available in the American market, only 13 had been given the "green light" by the Food and Drug Administration (FDA) (*Alugubelli, Abuissa & Roka, 2022*).

HRV measurement preparation involves determining the duration, ranging from 5 min for basic parameters to longer for long-term fluctuations and non-linear parameters. Individuals with high ectopic beats (beats originating outside of the sinoatrial node), exceeding 20–30%, are generally not suitable for HRV analysis, except in cases where HR turbulence analysis is being conducted. It is essential to address artefacts and identify and manage ectopic beats during the measurement process (*Johnston et al., 2020*). Moreover, maintaining a sample rate of at least 250 to 500 Hz, as recommended in the consensus report published in 1996, is vital to avoid significant distortion in HRV data (*Ernst, 2017*; *Task Force of the European Society of Cardiology the North American Society of Pacing Electrophysiology, 1996*).

To ensure consistency in physiological and clinical studies, the ESC/NASPE recommends two types of recordings: short-term recordings of 5 min under stable physiological conditions processed using frequency-domain methods, and nominal 24-hour recordings processed by time-domain methods. Both methods yield correlated data, but the time-domain analysis is considered technically simpler and less susceptible to interference, making it more practical for routine clinical use (*Stys & Stys, 1998*).

The analysis is obtained under controlled standardized conditions (*Cygankiewicz & Zareba, 2013*). Short-term HRV measurements come from two different but connected systems in our body. The first one involves how our PNS and SNS work together and constantly change. The second source is a combination of various mechanisms like the control of blood pressure through feedback, RSA, and rhythmic changes in how our blood vessels behave, all of which help control our HR (*Shaffer & Ginsberg, 2017*).

## HRV analysis techniques

This section describes the techniques used to analyze HRV in the existing literature. The two most widely used types of analysis carried out in HRV are time-domain analysis and frequency-domain analysis. These indices offer insights into the autonomic control of the heart and whether measured in time or frequency-domains, are interconnected and reveal the influence on heart modulation of the parasympathetic, combined sympathetic-parasympathetic, and circadian rhythms (*Stein & Kleiger, 1999*). Additionally to these techniques, other non-linear analysis techniques are being used by the experts. Furthermore, many studies combine time-domain and frequency-domain analyses to obtain a more comprehensive assessment of autonomic function. Table S2 links these HRV metrics to specific studies. The following subsections discuss the main characteristics of these analysis techniques and touch upon their physiological sources.

### *Time-domain analysis*

Time-domain methods encompass both statistical and geometric techniques. Seven statistical and four geometrical indices are mentioned in the consensus report by

**Table 1  Time-domain metrics for HRV analysis.**

| Variable | Units | Description |
|---|---|---|
| **Statistical measures** | | |
| SDNN | ms | Standard deviation of all NN intervals |
| SDANN | ms | Standard deviation of the averages of NN intervals in all 5-min segments of the entire recording |
| SDRR | ms | Standard deviation of all RR intervals |
| RMSSD | ms | Root mean square of the successive differences between adjacent NN intervals |
| SDNN index | ms | Mean of the standard deviations of all NN intervals for all 5-min segments of the entire recording |
| SDSD | ms | Standard deviation of differences between adjacent NN intervals |
| NN50 count | – | Number of pairs of adjacent NN intervals differing by more than 50 ms in the entire recording |
| pNN50 | % | NN50 count divided by the total number of all NN intervals |
| **Geometric measures** | | |
| HRV triangular index | – | Total number of all NN intervals divided by the height of the histogram of all NN intervals measured on a discrete scale with bins of 1/128s |
| TINN | ms | Baseline width of the minimum square difference triangular interpolation of the highest peak of the histogram of all NN intervals |
| Differential index | ms | Difference between the widths of the histogram of differences between adjacent NN intervals measured at selected heights |
| Logarithmic index | $ms^{-1}$ | Coefficient $\phi$ of the exponential curve, which is the best approximation of the histogram of absolute differences between adjacent NN intervals |

the NASPE/ESC. The following four measures for time-domain HRV analysis are recommended by NASPE/ESC: SDNN, standard deviation of the averages of NN intervals (SDANN), RMSSD, and HRV triangular index (*Cygankiewicz & Zareba, 2013*; *Task Force of the European Society of Cardiology the North American Society of Pacing Electrophysiology, 1996*). Table 1 shows the details of these time-domain metrics.

The measurement of these time-domain metrics was made possible by the development of computerized data storage and signal processing. Employing advanced signal processing techniques, the removal of ectopic beats from the ECG signal is made possible. An ECG signal consists of three main segments: the P segment, the QRS complex, and the T segment. In this type of filtration, the P segment of the ECG signal is inspected. The absence of the P segment indicates that the heartbeat did not originate at the sinoatrial (SA) node, and therefore, does not convey information about the ANS (*Draghici & Taylor, 2016*). Such beats are referred to as the ectopic beats. After performing ectopic filtration from the output ECG signal, RR intervals are converted into NN intervals (*Peltola, 2012*; *Cygankiewicz & Zareba, 2013*; *Johnston et al., 2020*). This filtering process ensures that HRV analysis only

**Table 2  Frequency-domain metrics for HRV analysis.**

| Variable | Units | Description | Frequency range |
|---|---|---|---|
| | | **Analysis of short-term recordings (5 min)** | |
| Total power | $ms^2$ | The variance of NN intervals over the temporal segment | $\approx \leq 0.4$ Hz |
| VLF | $ms^2$ | Power in very low frequency range | $\leq 0.04$ Hz |
| LF | $ms^2$ | Power in low frequency range | 0.04–0.15 Hz |
| LF norm | nu | Relative low frequency power in normalized units LF/(total power-VLF) $\times$ 100 | n/a |
| HF | $ms^2$ | Power in high frequency range | 0.15–0.4 Hz |
| HF norm | nu | Relative high frequency power in normalized units HF/(total power-VLF) $\times$ 100 | n/a |
| LF/HF | % | Ratio LF $(ms^2)$/HF$(ms^2)$ | n/a |
| | | **Analysis of entire 24 hours** | |
| Total power | $ms^2$ | Variance of all NN intervals | $\approx \leq 0.4$ Hz |
| ULF | $ms^2$ | Absolute power in the ultra-low frequency band range | $\leq 0.003$ Hz |
| VLF | $ms^2$ | Absolute power in the very low frequency range | 0.003–0.04 Hz |
| LF | $ms^2$ | Absolute power in the low frequency range | 0.04–0.15 Hz |
| HF | $ms^2$ | Absolute power in the high frequency range | 0.15–0.4 Hz |

considers heartbeats originating from the SA node (*Li, Rüdiger & Ziemssen, 2019*; *Stys & Stys, 1998*).

The choice of analysis duration also plays a crucial role in determining the physiological processes that a matric predominantly correlates with. Each of the time-domain metrics shown in Table 2 requires a specific time frame for measurement and reflects different aspects of HRV accordingly. Five-minute and twenty-four-hour recordings are suggested as standards for short-term and long-term measurements because the value of time-domain parameters strongly depends on the length of the analyzed time interval (*Hejjel & Gál, 2001*). RMSSD and NN50 count divided by the total number of all NN intervals (pNN50), for instance, are capable of capturing changes in HRV over relatively short periods, typically up to 5 min.

Regarding long-term measurements of SDNN, as evidenced by a substantial body of literature, often demand data collected over a more extended duration, such as 24 h. For example, over a two-minute interval, the SDNN primarily reflects RSA or PNS activity but for a 24-hour SDNN measurement slower processes, such as circadian rhythms or hormonal fluctuations, take centre stage. SDANN is the best measurement to reflect circadian variations in HRV.

In essence, the analysis of successive differences in NN intervals targets the faster components of heart rhythm, including high-frequency HRV and RSA, which are predominantly parasympathetic in nature. SDNN is thought to represent overall variability. There is not a single time-domain HRV parameter that could be said to primarily represent sympathetic heart modulation (*Cygankiewicz & Zareba, 2013*).

Although the strength of the correlation varies, the time-domain indices are positively correlated with one another (*Cowan, 1995*). The SDNN is found to be highly correlated with frequency-domain parameters such as ultra-low frequency (ULF) and total power.

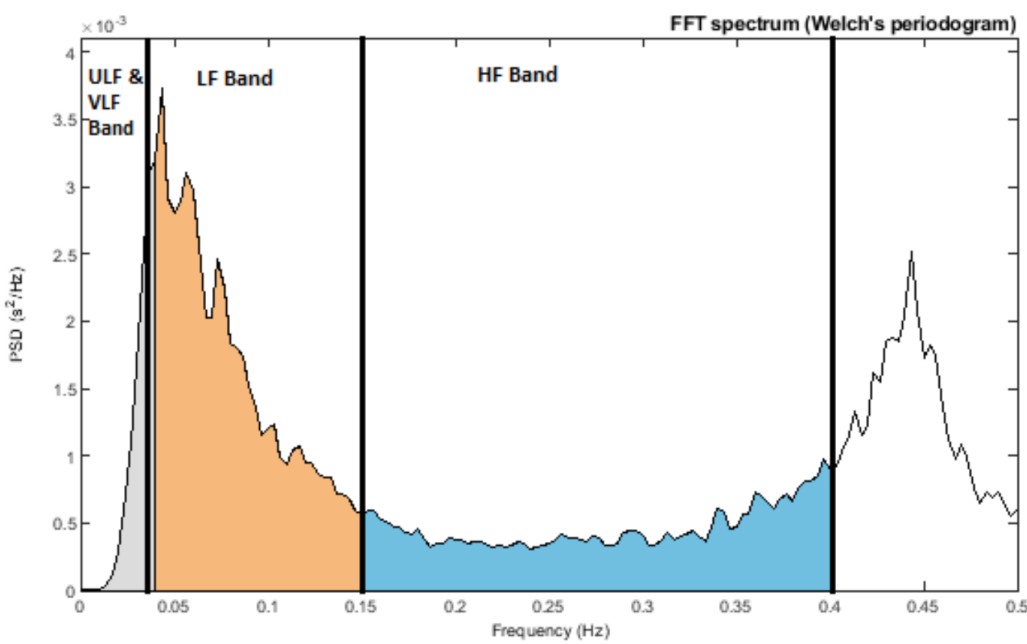

**Figure 4** Power spectral density of RR interval signal indicating ULF and VLF bands (≤0.04 Hz), LF Band (0.04–0.15 Hz), and HF Band (0.15–0.4 Hz).

Highly correlated time-domain metrics are SDNN and SDANN to RMSSD and pNN50, respectively (*Stein & Kleiger, 1999*; *Cowan, 1995*).

### *Frequency-domain analysis*

As early as the 1960s, investigators began to apply spectral analysis approaches to the RR series by using fast Fourier transform and autoregressive modelling. Similar to a prism dispersing white light into various colours of different wavelengths, one can think of the frequency-domain analysis as analogous to placing a photosensor in front of the dispersed light, which measures the strength of individual light components (*Bilchick & Berger, 2006*). Using frequency-domain analysis, we can quantify the spectral power for each frequency range. This spectral power is distributed according to frequency, and the higher the frequency, the faster the cycle. These frequency ranges are typically divided into several categories: high-frequency (HF), low-frequency (LF), very-low frequency (VLF), and ultra-low frequency (ULF). The specific frequency ranges are detailed in Table 2, and the power spectral density of an RR interval showing different frequency ranges is shown in Fig. 4.

HF is associated with RSA, whereas LF is linked to baroreflex activity and vascular sympathetic nerves (known as Mayer waves) (*Reyes del Paso et al., 2013*; *Kobayashi, Ishibashi & Noguchi, 1999*; *Kleiger, Stein & Bigger Jr, 2005*; *Draghici & Taylor, 2016*). VLF pertains to slower physiological systems, including temperature regulation and hormone activity. Furthermore, the heart itself possesses an intrinsic nervous system believed to operate in the VLF range (referred to as the heart's intrinsic nervous system) (*Kleiger, Stein & Bigger Jr, 2005*). Lastly, ULF involves processes such as thermoregulation and

the secretion of hormones like cortisol and growth hormones, which follow a circadian rhythm. The exact physiological mechanisms responsible for ULF and VLF have not yet been established (*Draghici & Taylor, 2016*). The physiological basis of ULF is debated, with strong correlations between it and vagal HRV markers and no correlations with sympathetic markers. HF reflects parasympathetic activity, whereas LF denotes both (*Hinde, White & Armstrong, 2021*). The LF/HF ratio is used to measure autonomic balance, with a low ratio indicating PNS dominance and a high ratio indicating SNS dominance. However, the precision of this ratio is debated and is discussed in detail in the contradiction section below (*Reyes del Paso et al., 2013*; *Nayak et al., 2023*; *Hayano & Yuda, 2021*).

Moreover, before conducting spectral analysis of HRV, a thorough examination of the ECG data is essential to detect and address potential problems such as artefacts, ectopic beats, and arrhythmic events. For short-term HRV analysis, it is advisable to select recordings free from such irregularities. However, if the data include technical artefacts like missed beats or electrical noise, these can be rectified through proper interpolation techniques based on the surrounding RR intervals. Managing ectopic beats is a more complex challenge, with various approaches available, including deletion or interpolation. Nevertheless, simply deleting ectopic beats is discouraged as it results in a loss of valuable ECG information and distorted LF and HF power. The choice of interpolation methods should be determined based on the specific nature of the ectopic beat, data quality, and the study population (*Li, Rüdiger & Ziemssen, 2019*; *Johnston et al., 2020*).

By breaking down the frequency into its components, one can learn more about the overall variability. This procedure allows for the quantification of the sympathovagal balance that regulates sinus node pacemaker activity under a variety of physiological and pathophysiological circumstances. Although changes in sympathovagal balance can frequently be detected under basal conditions, the most prevalent trait that distinguishes many pathophysiological states is a decreased responsiveness to an excitatory stimulus. Additionally, a diminished oscillatory pattern or one that is less responsive to a specific stimulus may also indicate a changed target function, providing intriguing prognostic indicators (*Montano et al., 2009*). The autorhythmicity of the sinoatrial node is influenced by various factors, adding variability to the HR signal (*Task Force of the European Society of Cardiology the North American Society of Pacing Electrophysiology, 1996*).

Spectral analysis of HRV in both short-term and long-term recordings also has advantages and disadvantages. In the case of short-term analysis, it is relatively easy to perform and offers convenient control over potential confounding factors. It requires less time for data processing and can effectively capture dynamic HRV changes within a short time frame. However, its disadvantage lies in its lack of stability due to the constant fluctuation of HR intervals, and it cannot analyze ULF power. On the other hand, long-term HRV measurement provides a stable platform for HRV analysis and the capacity to analyze ULF power. However, it comes with the drawback of being more expensive and time-consuming. Long-term recordings tend to include more noise and are influenced to a greater extent by various activities and environmental factors (*Li, Rüdiger & Ziemssen, 2019*).

Additionally, each of the 24-hour spectral metrics correlates to a time-based metric that is strongly connected to it. This correlation arises because both types of measures are affected by the same physiological factors and share mathematical relationships. The choice between using time-domain-based metrics like SDNN and frequency-domain metrics in a study depends on personal preference and available tools. Because some HRV data has an uneven distribution, it is common to use log transformations when doing statistical analyses. pNN50 and RMSSD are correlated with HF power (*Kleiger, Stein & Bigger Jr, 2005*; *Xhyheri et al., 2012*). In addition, SDNN correlates with total power (TP), SDANN with ULF, and SDNN index with both VLF and LF (*Stein & Kleiger, 1999*; *Cowan, 1995*).

### Non-linear analysis

Nonlinear methods are highly effective for analyzing HRV due to their ability to capture the complex, dynamic, and nonstationary nature of HR signals, which are influenced by various physiological factors. They provide a more accurate representation of HR dynamics compared to traditional linear approaches, particularly in the presence of noise and nonstationarity (*Rajendra Acharya et al., 2006*). Non-linear measurements explore the unpredictability of a time series, arising from the intricate and multifaceted mechanisms governing HRV. Nonlinear indices have particular relevance when they correlate with frequency-domain and time-domain metrics that stem from the same underlying processes. These non-linear approaches provide fresh and innovative perspectives for scrutinizing HRV data. Although they offer novel insights, it is worth noting that non-linear methods are not as commonly employed as time and frequency-domain analyses (*Ernst, 2017*). However, unlike time-domain and frequency-domain approaches, which have limitations such as an inability to differentiate sympathetic and parasympathetic activity or issues like spectral leakage, nonlinear methods overcome these shortcomings (*Rajendra Acharya et al., 2006*).

A few popular non-linear analysis methods include the Poincaré plot, detrended fluctuation analysis (DFA), entropy analysis, and recurrence plots (*Nayak et al., 2023*). Poincaré plots (scatter plots) are visual graphs displaying HRV by plotting each RR interval against the previous one. These plots help analyze HRV data by fitting an ellipse to the plotted points and deriving three non-linear measures: SD, SD1, and SD2. These measures represent the variability within the data. DFA assesses fluctuations in RR intervals across different time scales, distinguishing short-term from long-term fluctuations. DFA addresses signal non-stationarity by removing extrinsic fluctuations, but it requires extensive data. Poincaré plots and DFA reveals short- and long-term variability, offering insights into autonomic regulation over different time scales. Entropy analysis measures irregularity or randomness within RR interval series, calculating the probability of interval sequence repetition. This approaches helps in identifying complex patterns that may not be evident through linear analysis. Lower entropy values often correlate with health problems (*Johnston et al., 2020*; *Bilchick & Berger, 2006*; *Nayak et al., 2023*).

Approximate entropy (ApEN) was introduced in the early 1990s as a significant algorithm for HRV analysis. It evaluates data sets for repeating structures and the probability of finding similar patterns in different time periods. ApEN returns values between 0 and around 1,

where lower values indicate higher regularity. Sample entropy (SampEn) was developed as an alternative to ApEN due to concerns about the latter's internal consistency. SampEn calculates the predictability of finding specific matches in short time-series and requires fewer data points, making it applicable for shorter HRV measurements. Non-linear algorithms, including ApEN and SampEn, have been successfully applied in various clinical studies, such as examining HRV behavior in patients with depression or respiratory disorders. These methods are particularly valuable for detecting unknown relationships between seemingly unrelated systems (*Ernst, 2017*). For clinical applications, research suggests that nonlinear methods like ApEn and SampEn improve diagnostic capabilities by detecting subtle HR pattern changes often overlooked by traditional methods. They provide insights into autonomic dysfunction in depression and respiratory disorders and revealed complex physiological relationships, such as disrupted circadian rhythms. In athletic applications, nonlinear methods are used to monitor recovery, training status, and stress response, providing predictive insights into fatigue and overtraining to optimize performance and prevent injuries (*Nayak et al., 2023*).

Recurrence analysis helps identify repeating patterns in cardiac rhythms, providing measures like recurrence rate and determinism to assess heart stability and predictability.Recurrence plot quantification analysis (RQA) is a nonlinear analysis method that works well with short, noisy, and non stationary time series like HRV. It is based on the study of textures seen in recurrence plots. From each HRV time series' recurrence plot, the complexity measures laminarity (LAM) and determinism(DET) were computed. The ratio of recurrence points that create diagonal structures is known as DET. Laminar states in the system are represented by LAM, which is defined as the ratio between the total set of recurrence points and the recurrence points that constitute the vertical structures. By incorporating principles from chaos theory, nonlinear methods allow researchers to examine the chaotic nature of heart rhythms, enhancing the understanding of physiological complexity and health status. These methods improve diagnostic potential for conditions like cardiac arrhythmias and other cardiovascular diseases (*Nayak et al., 2023*). Table 3 shows details about non-linear metrics for HRV analysis.

## Confounding factors

It is important to recognize that HRV is influenced by a multitude of confounding factors, which can be categorized into various groups: physiological factors, pathological factors, environmental factors, lifestyle factors, psychological factors and non-modifiable factors. Physiological factors comprise aspects such as age, sex, circadian rhythms, physical activity, and respiration patterns, all of which play a significant role in HRV regulation (*Li, Rüdiger & Ziemssen, 2019*). Pathological factors involve conditions like inflammation, infection, and sepsis, which can disrupt HRV patterns. Environmental factors include elements like social stressors, noise pollution, and exposure to toxins like carbon monoxide. Lifestyle factors encompass choices such as diet, alcohol consumption, and cigarette smoking, which can impact HRV. Lastly, psychological factors, such as mental stress, depression, and anxiety disorders, have a considerable influence on HRV patterns. Understanding the intricate interplay of these factors is crucial for accurately interpreting HRV data and its

**Table 3  Non-linear metrics for HRV analysis.**

| Non-Linear HRV analysis | Parameters | Units | Description |
| --- | --- | --- | --- |
| **Poincaré analysis** | S | ms | Area of the ellipse, representing total variability |
| | SD1 | ms | Standard deviation perpendicular to the line of identity |
| | SD2 | ms | Standard deviation along the line of identity |
| | SD1/SD2 | % | Ratio of SD1 to SD2 |
| **Entropy analysis** | ApEn | n/a | Approximate entropy, measures the regularity and complexity associated with a time series |
| | SampEn | n/a | Sample entropy, measures the regularity and complexity associated with a time series |
| **Detrended fluctuation analysis** | DFA $\alpha1$ | n/a | Describes short-term fluctuations (less than 11 beats) |
| | DFA $\alpha2$ | n/a | Describes long-term fluctuations (more than 10 beats) |
| | D2 | n/a | Estimates the minimum number of variables required to model system dynamics |

implications for overall health and well-being (*Tiwari et al., 2021*). In future studies, in addition to ethnic variances and lifestyle elements, it is crucial to consistently consider factors like age, sex, body posture, and respiratory patterns when assessing HR and HRV (*Parati & Di Rienzo, 2003*).

Figure 5 displays various confounding factors categorized into their respective groups. This figure draws inspiration from the influence diagram in *Fatisson, Oswald & Lalonde (2016)*. The diagram uses different colours to indicate the type of effect on HRV values: red signifies harmful effects, green and blue indicate beneficial effects (with blue being specific to the heart coherence state), purple can be either beneficial or harmful depending on the environment, grey represents factors for which statistical significance has not yet been established and white shows the factors for which according to the best of our knowledge statistical significance has not been proven. Additionally, the figure illustrates indirect effects and influences through dotted lines, such as the impact of posture, sex, and age on the respiration rate, and the connections between the central nervous system and both physiological and neuro-psychological factors. These categories of confounding factors are shown to be interdependent.

A fundamental relationship exists between HRV and HR which can be both physiological and mathematical. The physiological HRV-HR relationship is determined by ANS activity. Higher PNS activity slows down HR, thus HRV increases and vice versa. The mathematical HRV-HR relationship is the nonlinear (inverse) relationship between RR interval and HR. HR and RR intervals are reciprocals of each other, or, to be exact, $HR = 60,000/RR$, where HR has units of beats per minute (bpm), and RR has units of milliseconds (ms) (*Bilchick & Berger, 2006*). Essentially, HRV is intricately linked to HR, where HRV tends to increase as the RR interval lengthens, signifying a slower HR. Conversely, as the RR interval shortens, indicating a quicker HR, HRV tends to decrease.

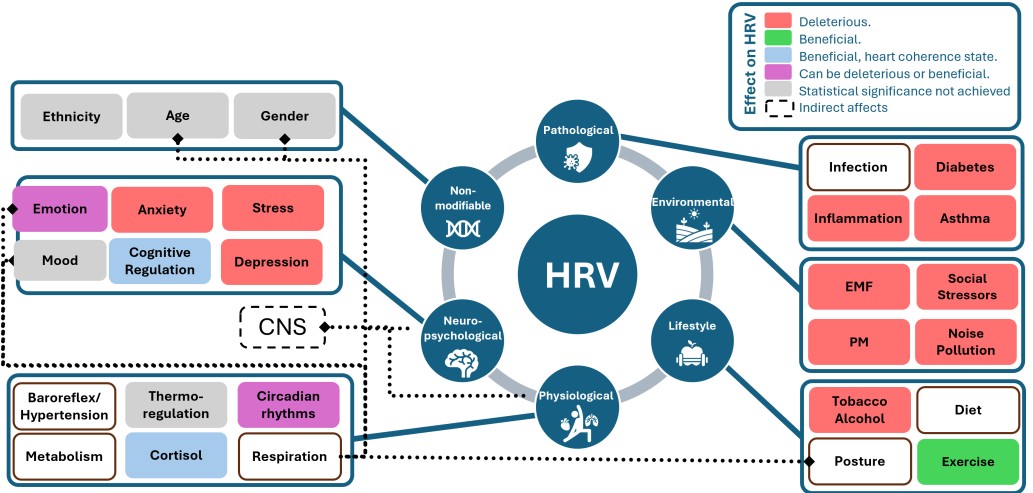

**Figure 5** Influence diagram of confounding factors linked with HRV, inspired from **Fig. 1** in the study by *Fatisson, Oswald & Lalonde (2016)*.

Another prominent contributor to HRV is the RSA, which manifests as fluctuations in HR during the respiratory cycle. During inhalation, the HR increases, and during exhalation, it decreases. This rhythmic pattern aligns with the typical respiratory cycle, spanning approximately 3 to 7 s per breath or about 9 to 24 cycles per minute. One more pivotal physiological factor impacting HRV is the baroreflex, a vital regulatory mechanism for maintaining blood pressure stability. The baroreflex operates on a slightly longer time scale, spanning approximately 3 to 20 s, or roughly 6 cycles per minute, equating to one adjustment every 10 s. When the PNS activates and the SNS inhibits, blood pressure increases, leading to a decrease in HR. Conversely, a decrease in blood pressure triggers PNS inhibition and SNS activation, increasing HR. RSA, and the baroreflex, are integral to understanding the dynamics of HRV, offering insights into the complex interplay between the respiratory cycle, blood pressure regulation, and HR variations (*Johnston et al., 2020*).

## HRV in clinical medicine

HRV faces challenges in gaining widespread acceptance in modern clinical medicine. Early ECG devices were heavy, and even wearable monitors in the late 1990s and 2000s posed inconveniences. A further constraint on HRV spectral analysis was the absence of standardization among the many computer approaches used by commercial Holter systems (this hinders relevant data comparisons *Lombardi, 2002*; *Stys & Stys, 1998*). However, recent advancements in wearable technology have improved data collection. Despite these advances, integrating ECG data into electronic medical records remains complex due to the diversity of record systems. HRV's focus on overall health rather than disease-specific diagnosis further complicates its fit within conventional healthcare models. Insurance coverage for HRV-related services is also often lacking, and the complexity of data interpretation adds to this challenge (*Gupta, Mahmoud & Massoomi, 2022*).

One of the main reasons why HRV is not being used in a clinical setting seems to be because the HRV data can be too messy and has several confounding factors. Another reason is that HRV is thought to have low specificity, sensitivity, and a low positive predictive value (*Kleiger, Stein & Bigger Jr, 2005*). Also, HRV does not have a reliable value for reference; it is sort of a positive health marker. Even in the late 1990s, there was no consensus about the best available index of HRV for clinical use (*Huikuri et al., 1999*). HRV analysis is widely studied in cardiac and non-cardiac diseases, but it has not been routinely integrated into diagnostic and prognostic processes due to the lack of accredited reference values for normal subjects and the changing profiles of patients, necessitating continuous updates of pathological thresholds. Establishing a clear distinction between what is considered normal and abnormal in HRV values can be subjective, making it demanding to effectively apply HRV in clinical practice (*Xhyheri et al., 2012*).

Even so, although HRV has been applied to a wide range of cardiovascular diseases, a consensus has only been reached so far concerning two clinical conditions. The 1996 consensus report identified two distinct clinical scenarios in which HRV analysis should be carried out: to identify early signs of cardiac autonomic neuropathy in diabetic patients and to assess mortality risk in post-myocardial infarction patients (*Task Force of the European Society of Cardiology the North American Society of Pacing Electrophysiology, 1996*). Studies have demonstrated that HRV is a predictor of post-myocardial infarction mortality, showing that reduced SDNN values are associated with a significantly higher risk of death. Similarly, HRV is useful in detecting diabetic autonomic neuropathy, as spectral analysis reveals autonomic dysfunction even before clinical symptoms appear (*Van Ravenswaaij-Arts et al., 1993*; *Bilchick & Berger, 2006*; *Johnston et al., 2020*). However, clinical use of HRV is still limited (*Cygankiewicz & Zareba, 2013*; *Ernst, 2017*; *Xhyheri et al., 2012*; *Task Force of the European Society of Cardiology the North American Society of Pacing Electrophysiology, 1996*). Before improved criteria can be defined for current and future therapeutic applications, more research must be done to determine the precise sensitivity, specificity, and predictive value of HRV as well as the normal levels of standard measurements in the general population (*Stys & Stys, 1998*). Because of the complicated methodology and lack of standardization, its clinical usefulness remains limited.

Additionally, when clinicians adopt pharmacological interventions for patients suffering from ischemic cardiac diseases, as these drugs often modify the heart rhythm, HRV is even more difficult to interpret. However, we can count on research that has shown that HRV can help predict outcomes in different clinical settings (*Lombardi, 2002*). On the other hand, while physiological studies primarily suggest recording heart rhythm under tightly controlled circumstances to reduce any potential interfering variables, this is challenging to perform in clinical practice (*Kautzner & John Camm, 1997*). Several steps and precautions are to be taken into account during measuring HRV clinically, specifically a 24-hour ECG reading. According to the study (*Catai et al., 2020*), before collecting HRV, a clinical evaluation of the subjects is necessary, accompanied by a basal electrocardiogram and a careful interpretation of it. Precautionary measures include a 30-question checklist, alongside checking data and analyzing demographic factors such as body mass, height, age, and sex. A careful recording of any drug used at that time is also mandatory.

Moreover, it involves setting up the data collection environment by assessing noise levels, temperature, humidity, and the time of day, as well as monitoring the flow of individuals entering or exiting the environment. It is important to ensure that the patient is aware of his/her surroundings and the HRV measurement apparatus. Patient preparation also includes refraining from consuming stimulant beverages, alcohol intake, smoking or engaging in high-intensity physical activity before the session, as well as abstaining from mobile phone usage. During HRV measurement, factors such as the patient's activity state (resting, exercising, sitting, or lying down) should be documented, and both the patient and observer must not speak during recording any instances of sneezing or coughing. Additionally, monitoring breathing rate, blood pressure, and HR is essential for the accuracy of the process.

In the 1990s, even though there was a large body of literature concerning HRV in cardiology and medical journals as shown in Fig. 3, its measurement from the Holter recordings had not yet become a routine clinical tool (*Huikuri et al., 1999*). This decade also saw increased interest in studying the ANS and its connection to cardiovascular health. This heightened attention was due to the noninvasive nature of HRV measurement for assessing the ANS. Unfortunately, the increased availability of HRV metrics led some researchers to use them extensively without a focused research plan. This lack of specificity can result in the identification of false patterns or leads that may not have real significance, requiring additional effort to correct and interpret the findings accurately. Consequently, it became challenging for clinicians to navigate this field and grasp the clinical significance of HRV, highlighting the importance of more targeted and hypothesis-driven research in its application (*Kautzner & John Camm, 1997*). A lack of complete understanding of the physiological significance of HRV parameters has hindered the wider and clearer use of this method (*Lombardi, 2002*).

Moreover, physicians encounter various obstacles when considering the integration of PPG-based wearable/device data into clinical practice. Firstly, their lack of familiarity with these devices, due to limited formal training, hampers their confidence in utilizing and interpreting the data. Second, concerns about data accuracy, including measurements such as HR or sleep patterns, contribute to scepticism regarding clinical decision-making based on this information. Also, the scarcity of robust clinical trials validating PPG-based wearable/device data for improving patient outcomes is a significant challenge for evidence-based medical practices. Time constraints during clinical encounters further impede the incorporation of PPG-based wearable/device data analysis, potentially disrupting workflow and patient interaction. Additionally, the multitude of available devices and varied health metrics they collect pose standardization challenges, compounded by the lack of consensus on interpreting HRV data derived from these devices, which adds complexity to clinical utilization. Overcoming these obstacles will require ongoing education, research, and standardization efforts to ensure that smartwatch and commercially-available device data can be effectively integrated into clinical practice in a meaningful way (*Gupta, Mahmoud & Massoomi, 2022*; *Tiwari et al., 2021*).

AI-driven tools, especially deep learning models like artificial neural network (ANNs) and convolution neural networks (CNNs), have increasingly been applied since 2018,

demonstrating high accuracy in HRV analysis, with CNNs achieving over 99.4% accuracy. While cardiology applications show strong predictive performance with low variance, fields like addiction and mental health require further validation due to high variability. Future research should focus on standardization efforts, refining AI methodologies, and conducting large-scale clinical trials to validate wearable technology for broader clinical adoption (*Faust et al., 2022*).

## Contradictions in the literature

When reading the literature, we also came across some contradictions with the HRV frequency-domain values like the LF values, or the ratio between HF and LF being the sympathovagal balance. Although it might be tempting to think of the ANS as a simple on/off switch, the reality is far more intricate. The ANS functions as a dynamic system; it does not operate in a binary manner. Instead, imagine it as a dial that can be turned up or down. The SNS and PNS are in a constant dance with each other within our bodies, modifying their influence based on our physiological and psychological needs. Rarely does one entirely dominate the other; a delicate balance is maintained.

The frequency-domain metric LF/HF ratio was initially thought to reflect the balance between parasympathetic and sympathetic activity but has been questioned. This ratio has been traditionally viewed as a representative measure of the balance between the PNS and the SNS (*Hinde, White & Armstrong, 2021*). It assumes that the PNS creates HF power, whereas the SNS might produce LF power. When the LF/HF ratio is low, it suggests that the PNS is in control, which is linked to relaxation. Conversely, when we experience stress or a fight-or-flight response, the LF/HF ratio is high, indicating that the SNS is dominant (*Lombardi et al., 1996*). However, this idea is debated for a couple of reasons. First, LF power does not only represent SNS activity; the PNS also plays a significant role in this frequency range, and there are some unknown factors involved. Second, the interactions between the PNS and SNS are very complex, nonlinear, and not always balanced, making it challenging to rely solely on this ratio for a complete understanding of our body's responses (*Shaffer & Ginsberg, 2017*; *Reyes del Paso et al., 2013*; *Nayak et al., 2023*; *Hayano & Yuda, 2021*; *Cygankiewicz & Zareba, 2013*; *Stein & Kleiger, 1999*). There is controversy in the literature about the physiological source of LF. In the beginning, it was recorded that there is more LF when there is more physical activity or more sympathetic response, so it became a way to measure or visualize the sympathetic response, but when they did short-term resting recordings of HRV, it showed no significant sympathetic activity. It may still provide useful insights in specific contexts, such as when subjects are at rest, in controlled settings, or when used for preliminary assessments in defined populations (*Schäfer & Vagedes, 2013*).

Another contradiction arose when the use of wearable devices with PPG sensors made it easier to measure HRV in everyday life. Many studies try to use PRV as a replacement for HRV (*Schäfer & Vagedes, 2013*). However, there are challenges in applying the traditional approach to PRV in daily activities. PRV is not just a copy of HRV; it is a different marker. The process from the heart's electrical signal (ECG R wave) to the pulse wave (PPG wave) involves many steps, and various factors can affect each step. Evidence shows differences

**Table 4** Summary of contradictions in HRV research and their implications.

| Contradiction | Summary | Practical implications |
|---|---|---|
| **LF/HF ratio** | Does it reflect sympathovagal balance? | Useful in controlled settings but limited by nonlinear ANS interactions and requires cautious interpretation. |
| **PRV *vs.* HRV** | Can PRV replace HRV? | PRV is practical for daily monitoring but may overestimate HRV in some contexts. |
| **HR-HRV relationship** | Hyperbolic *vs.* negative exponential models. | Impacts interpretation of HRV metrics, especially in clinical applications. |
| **HRV terminology** | Should HRV be called "heart period variability"? | Highlights the need for precise terminology to avoid confusion in research. |

between PRV and HRV, like PRV being present even when HRV is not. Additionally, there are variations in HRV and PRV values depending on the position of the sensor/electrodes, as well as different effects due to posture and exercise. Also, the PPG approach overestimates the HRV in comparison to the ECG. Understanding these differences is essential for using these measures accurately in real-life situations (*Hayano & Yuda, 2021*; *Alugubelli, Abuissa & Roka, 2022*).

Another phenomenon currently under debate is related to the relationship between HRV and HR. The literature is divided regarding the mathematical nature of this relationship, with some proposing it behaves like a hyperbolic function, whereas others suggest it resembles a negative exponential relationship. The hyperbolic model suggests that HRV decreases proportionally with increasing HR, while the negative exponential model implies a more rapid decline in HRV at higher HR levels (*Draghici & Taylor, 2016*; *De Geus et al., 2019*). In addition, there is another complaint that technically, HRV should be called heart period variability because it measures the time between heartbeats, not the number of heartbeats per minute(HR) (*De Geus et al., 2019*). Table 4 provides a summary of contradictions in HRV research and their implications.

## Potential of HRV

Although it is not yet widely used in the medical community, athletes and entrepreneurs in the technology sector who are seeking to optimize physical and cognitive performance have found HRV increasingly relevant. Recent studies have revealed that wearable devices reported small error rates when compared to ECG, and the practicality and accessibility of the wearable devices outweigh these negligible errors (*Yuda et al., 2020*; *Schäfer & Vagedes, 2013*; *Hinde, White & Armstrong, 2021*; *Dobbs et al., 2019*). Devices like the Apple Watch, Fitbit Charge, Garmin VivoSmart, and Polar A360 wrist monitors are finding utility not only in cardiology, sleep medicine, and diabetes detection and management but also in the rising interest of individuals using remote HR monitors. There are currently over 1,000 clinical trials underway in this field. As more data becomes available, the validity of consumer-grade wearable monitors for HRV-based applications will become clearer (*Alugubelli, Abuissa & Roka, 2022*).

Furthermore, HRV's uniqueness to each individual makes it an excellent candidate for personalized models (*Hinde, White & Armstrong, 2021*). Monitoring HRV can aid

in prescribing individualized training regimens, helping determine training session intensity based on an individual's PNS activity compared to their previous measurements. HRV-guided training has proven to be a beneficial tool for enhancing endurance running performance (*Singh et al., 2018*; *Alugubelli, Abuissa & Roka, 2022*; *Hinde, White & Armstrong, 2021*).

The realm of mHealth, involving the use of mobile devices to support medical care and public health, is evolving rapidly and can be challenging to navigate. This encompasses mobile applications, wearable sensors, and communication technologies. Numerous studies support the validity of PRV compared to the ECG gold standard, which applies to both healthy adults and athletes. Establishing normal and abnormal values is complex, requiring consideration of age and sex, and current studies lack established norms. However, device manufacturers provide data indicating average values among users. The widespread adoption of PPG-based wearable devices by patients without clinician consultation underscores the need for healthcare professionals to embrace and leverage the valuable data offered by these commonly encountered devices (*Gupta, Mahmoud & Massoomi, 2022*). Also, integrating HRV assessment into trauma care has the potential to enhance the precision and effectiveness of patient management strategies, contributing to improved outcomes in critical injury cases (*Tiwari et al., 2021*).

The evidence supporting machine learning-based HR and HRV analysis is rapidly accumulating, with a surge in the number of studies utilizing artificial intelligence in recent years. Several promising machine learning and deep learning algorithms have demonstrated their potential for HR/HRV analysis. Prospective studies will play a crucial role in assessing the clinical and non-clinical utility of these approaches. Additionally, machine learning-based analysis offers the potential to alleviate the burden on healthcare systems (*Alugubelli, Abuissa & Roka, 2022*; *Ishaque, Khan & Krishnan, 2021*). Early applications of machine learning in HRV analysis have been tailored for specific diagnostic purposes, such as the detection of obstructive sleep apnea (achieving 93% accuracy using support vector machines (SVM)), congestive heart failure (achieving up to 99% accuracy using SVM), and diabetes mellitus (with an average accuracy of 90% using adaBoost, decision trees, feature selection, k-nearest neighbors, probabilistic neural networks, and SVM). Mobile applications now use machine learning algorithms to provide real-time HRV insights, enabling users to monitor their autonomic health dynamically. AI-driven apps can detect anomalies in HRV patterns, offering personalized recommendations for stress management or recovery. Exploring these intersections between mHealth and AI can significantly enhance real-time HRV analysis by improving accuracy, user engagement, and accessibility, while also enabling automated and personalized health insights in real time (*Alugubelli, Abuissa & Roka, 2022*).

Medical decision support can be used to transfer the analytical work from humans to computers, thereby enhancing healthcare systems. HRV-based medical decision support can be delivered through three methods: directly tracking patient health using extracted features, combining multiple features with machine learning algorithms to enhance decision quality, and using deep learning for automatic labelling of HR signal samples without explicit feature engineering, thereby providing a direct and independent

information pathway for decision support from the patient to the physician (*Faust et al., 2022*; *Alugubelli, Abuissa & Roka, 2022*).

## DISCUSSION

The difficulty in comprehending how the heart operates and its interconnectedness with the brain and other parts of the body is a challenging task. HRV is described as "one of the most complex and variable terms" associated with the heart. This implies that HRV is not a single static measurement but rather a multifaceted parameter influenced by numerous factors. The generation of each heartbeat involves various complex components of the heart and connections from the brain. Both the physical state and mental state of an individual, as well as environmental factors, can impact HRV. All these convoluted factors interact with each other and collectively contribute to variations in the intervals between heartbeats. Since HR and HRV values can be measured, they provide a means to gain insight into the complex environment surrounding the human heart. In other words, by quantifying HRV, researchers and healthcare professionals can attempt to understand the interplay of these complex factors

Recently, there has been a surge in the manufacturing of consumer-grade devices for measuring HRV. To understand whether these consumer-grade heart monitoring devices are good for tasks like measuring HRV, we need more information and research. As more studies are done and more data is collected, we will have a better idea of how reliable and useful these devices are for healthcare purposes.

As technology improves, it is becoming easier to keep an eye on people's health, especially their hearts, from a distance. Apart from combining PPG with accelerometry, advancements in artificial intelligence, machine learning, and deep learning algorithms, and high-performance miniature electronic sensors have enabled the development of modern wearable technology (*Chiang & Khosla, 2023*). Newer sensors and electrodes are less intrusive and can be integrated into clothing. Although consumer-grade wearable devices have shown accurate HR measurements, there is still a lack of robust data supporting their use for clinical HRV analysis. Further studies are essential to identify optimal software and techniques for artefact correction in RR signals, ensuring accurate HRV-derived parameters.

Data security remains a significant concern, especially regarding data storage and transmission. Remote monitoring can ease logistical burdens and detect potential problems, but it requires careful data management and patient-healthcare provider communication. Incorporating artificial intelligence into data processing shows potential but needs validation through large-scale studies. While machine learning-based HRV analysis holds promise, more prospective research is needed to fully understand its clinical implications. Further studies on sensor technologies and methodology for large-scale trials will clarify the strengths and limitations of these evolving technologies in real-world applications (*Alugubelli, Abuissa & Roka, 2022*).

Standardized recommendations are crucial for both clinicians and researchers. To adequately quantify and interpret HRV-derived parameters from short-term recordings

under resting conditions, attention to methodological details is imperative (*Plaza-Florido, Sacha & Alcantara, 2021*). Establishing standardized procedures will enhance the reliability and consistency of HRV assessments, benefiting both clinical practice and research endeavours.

Also, the ongoing contradictions surrounding HRV metrics, such as the controversy surrounding the LF/HF ratio, has significant implications for research and clinical use. In research, the lack of standardization of the LF/HF interpretation as an index of sympathovagal balance has contributed to inconsistencies in study results to the point that standardized conclusions are no longer easily transferrable across populations and conditions. In clinical practice, this lack of consensus poses challenges in integrating HRV into routine patient assessments. Physicians may be hesitant to use HRV measures in clinical decision making due to uncertainty about data reliability, which means inconsistent application of HRV-based risk stratification. Additionally, differences in methodologies can affect the reproducibility of HRV-based research results, limiting their applicability to evidence-based medicine. The resolution of these disparities with standardized protocols, optimized measurement guidelines, and validation in diverse clinical populations is necessary to improve the validity and utility of HRV analysis for clinical and research use.

Exploring these reviews revealed significant gaps in HRV research. We discovered a lack of literature providing consistent HRV values across all HRV matrices, highlighting the need for further studies to establish these baseline values. The future studies of HRV will benefit research and clinical use in the establishment of baseline values in more diverse populations, considering age, sex, and health conditions for better standardization. Additionally, given the significant changes in environmental factors and overall health since the 1990s, it is crucial to establish a protocol to account for potential confounding factors. For this future longitudinal research should look into how these changes affect HRV and PRV metrics. Similarly, as advanced sensor technology has evolved, there is a need for a protocol to define PRV values. Further research is necessary to understand the physiological reasons behind the discrepancies between HRV and PRV values. Addressing these research gaps will enable the determination of appropriate HRV metrics, potentially leading to the development of a new consensus report on HRV and PRV. Finally, research should concentrate on identifying specific HRV indicators with the most clinical utility, especially for risk stratification, stress monitoring, and cardiovascular health evaluation.

## CONCLUSIONS

This literature review aims to provide an understanding of HRV by presenting the evolutionary journey of this physiological quantity over the last five decades and an assessment of our current knowledge of the potential reasons why abnormal HRV may alleviate health problems. A better understanding of the mechanisms linking abnormal HRV to increased health problems might lead to specific therapeutic strategies that may reduce the risk of future complications for people with health problems. Moreover, an appraisal of the clinical applicability of the latest HRV measurement tools and the proposal of future directions for HRV research is part of this review. Understanding the physiological

discrepancies between HRV and PRV values is crucial for determining appropriate metrics and consensus reports. It is concluded that further research into the clinical applications and potential of wearable technology with advanced artificial intelligence-based signal processing for HRV is needed.

### Funding
The authors received no funding for this work.

### Competing Interests
The authors declare there are no competing interests.

### Author Contributions
- Amina Sundas conceived and designed the experiments, performed the experiments, analyzed the data, prepared figures and/or tables, authored or reviewed drafts of the article, and approved the final draft.
- Ivan Contreras conceived and designed the experiments, performed the experiments, analyzed the data, prepared figures and/or tables, authored or reviewed drafts of the article, and approved the final draft.
- Judith Navarro-Otano analyzed the data, authored or reviewed drafts of the article, provided clinical perspective, and approved the final draft.
- Júlia Soler analyzed the data, authored or reviewed drafts of the article, and approved the final draft.
- Aleix Beneyto analyzed the data, authored or reviewed drafts of the article, and approved the final draft.
- Josep Vehi conceived and designed the experiments, authored or reviewed drafts of the article, and approved the final draft.

### Data Availability
The reviewed articles and devices are listed as tables in a Supplementary File as Tables S1 and S2. This is a literature review hence it did not utilize raw data/code.

### Supplemental Information
Supplemental information for this article can be found online at http://dx.doi.org/10.7717/peerj.19347#supplemental-information.

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
