# Peer review of "Heart rate variability over the decades: a scoping review"

_PeerJ, doi:10.7717/peerj.19347_

## Round 0.1 · original submission · Major Revisions

Dear Authors, the manuscript is interesting and covers relevant topic in HRV research but, as suggested also by reviewers, needs several changes and improvements to be considered for a publication. I suggest to carefuly read and addressed all the comments provided by the reviewers.

Reviewer 1 ·

Basic reporting

Author Names: Please note that some of the author names are not properly capitalized (e.g., "amina sundas"). It is important to adhere to standard capitalization conventions for names to maintain professionalism and consistency. Kindly ensure all author names are capitalized appropriately.

Abstract: The abstract provides a clear overview of the study's scope, methodology, and findings, effectively highlighting the importance of HRV in assessing cardiac autonomic function and its clinical applications. However, there are areas for improvement. First, common abbreviations such as ANS (autonomic nervous system) and HR (heart rate), which are used in the main text, should be introduced in the abstract for clarity and consistency. Second, the abstract's focus could be refined to avoid repetition and better emphasize the study's unique contributions. For example, the absence of standardized protocols for HRV measurements is mentioned multiple times, which could be consolidated into a single, concise statement. Additionally, more emphasis on the key findings and novel insights from the review would provide better balance between discussing limitations and showcasing contributions. Overall, addressing these points would improve the clarity, focus, and impact of the abstract, ensuring it aligns with the professional standards of an academic journal.

Introduction: The introduction provides a thorough and detailed explanation of heart rate variability (HRV), its physiological basis, and its clinical significance. It effectively introduces the autonomic nervous system (ANS) and its components, including the sympathetic nervous system (SNS) and parasympathetic nervous system (PNS), while connecting these to HRV regulation. However, several enhancements could improve the clarity, focus, and impact of the introduction. Below are specific points for improvement and relevant references to support the arguments:

Abbreviation Clarity: Abbreviations such as ANS, SNS, and PNS should be defined at their first mention to ensure clarity for all readers.

Suggested Reference:
Gabella, G. (1976). Structure of the autonomic nervous system. Dordrecht: Springer. https://doi.org/10.1007/978-94-009-5745-9

Complexity of Heart Function: The introduction should emphasize the complexity of heart function. For instance, the irregular heartbeat observed in healthy individuals results from fluctuations in arterial blood pressure and stroke volume (Appel et al., 1989). Additionally, emotional stimuli, such as joy or anxiety, can influence HR differently depending on the context (Kreibig, 2010). Temporal variations and the intensity of the pulse wave can further affect HRV (Watanabe et al., 2022; Lim, 2022). Including these aspects will provide a more comprehensive view of HRV as an indicator of systemic health.

Suggested References:
Appel, M. L., Berger, R. D., Saul, J. P., Smith, J. M., & Cohen, R. J. (1989). Beat-to-beat variability in cardiovascular variables: Noise or music? Journal of the American College of Cardiology, 14(5), 1139–1148. https://doi.org/10.1016/0735-1097(89)90408-7
Kreibig, S. D. (2010). Autonomic nervous system activity in emotion: A review. Biological Psychology, 84(3), 394–421. https://doi.org/10.1016/j.biopsycho.2010.03.010
Watanabe, T., Hoshide, S., & Kario, K. (2022). Home blood pressure variability. Current Hypertension Reports, 24, 11–21. https://doi.org/10.1007/s11906-021-01168-5
Lim, C. Y. (2022). Variability in pulse waveforms and its implications for cardiovascular health. Frontiers in Physiology, 13, Article 873690. https://doi.org/10.3389/fphys.2022.873690

Repetitiveness and Streamlining: Some sections, such as the discussion on vagal and sympathetic influences, are repetitive and could be condensed. This will help maintain focus and reduce unnecessary detail, such as specific intrinsic heart rate values or modulation ranges.

Missing References: Certain statements, such as "HRV is generally regarded as a more sensitive index of the ANS than HR analysis," would benefit from additional citations to foundational or recent studies. Similarly, the explanation of respiratory sinus arrhythmia (RSA) and its contribution to HRV would be strengthened by references to studies that have investigated RSA.

Suggested References:
Litviňuková, M., et al. (2020). Cells of the adult human heart. Nature, 588(7838), 466–472. https://doi.org/10.1038/s41586-020-2797-4
Nanchen, D. (2018). Resting heart rate: What is normal? Heart, 104(13), 1048–1049. https://doi.org/10.1136/heartjnl-2017-312731
Olshansky, B., et al. (2017). Arrhythmia essentials (2nd ed.). Philadelphia: Elsevier.

Technological Advancements in HR Measurement: The introduction could benefit from discussing alternative HR measurement techniques, such as chest straps, smart clothing, intelligent speakers, and ear-canal sensors. Highlighting these advancements would demonstrate the evolving technological landscape of HR monitoring.

Suggested References:
Sartor, F., et al. (2018). Wrist-worn optical and chest strap heart rate comparison in a heterogeneous sample of healthy individuals and coronary artery disease patients. BMC Sports Science, Medicine and Rehabilitation, 10(1), 10. https://doi.org/10.1186/s13102-018-0098-0
Stangl, F. J., & Riedl, R. (2022a). Measurement of heart rate and heart rate variability with wearable devices: A systematic review. Proceedings of the 17th International Conference on Wirtschaftsinformatik.
Stangl, F. J., & Riedl, R. (2022b). Measurement of heart rate and heart rate variability: A review of NeuroIS research with a focus on applied methods. Information Systems and Neuroscience: NeuroIS Retreat 2022. Springer. https://doi.org/10.1007/978-3-031-13064-9_28

By addressing these points and incorporating the suggested references, the introduction will provide a more balanced, well-supported, and impactful narrative, setting a solid foundation for the rest of the paper.

Experimental design

The study design is well-structured and transparent, detailing the steps taken to conduct a scoping umbrella review of HRV literature. The use of multiple databases (PubMed, Web of Science, and Scopus) ensures comprehensive coverage, and the inclusion of a PRISMA methodology flow diagram adds credibility. However, a few areas could benefit from clarification and elaboration. First, the term "scoping umbrella review" should be defined to distinguish it from other types of reviews. Second, while the exclusion criteria are listed, providing a rationale for these choices (e.g., why therapeutic or gene studies were excluded) would help readers understand the scope of the review. Including an example of the search query used and a more detailed breakdown of the screening process (e.g., reasons for exclusion at each stage) would enhance reproducibility and depth. Additionally, the methodology should address how the accuracy of device information sourced from websites was ensured, given the potential for updates over time. Finally, the decision to present this as a scoping review rather than a systematic review should be justified in more detail to strengthen the methodological narrative.

Suggested References:
Paré, G., Trudel, M.-C., Jaana, M., & Kitsiou, S. (2015). Synthesizing information systems knowledge: A typology of literature reviews. Information & Management, 52(2), 183–199.
Schryen, G., Wagner, G., Benlian, A., & Paré, G. (2020). A knowledge development perspective on literature reviews: Validation of a new typology in the IS field. Communications of the Association for Information Systems, 46(7), 134–186.

Moreover, adding a clear research question or set of research questions would significantly enhance the focus and purpose of the study. Based on the content of the study design section, the authors could frame their research question as follows: What are the main measurement methods used for measuring heart rate variability (HRV), and how have they evolved over the past 50 years?
This research question would align with the stated goals of the review and provide a clear framework for readers to understand the scope and contributions of the paper. Encouraging the authors to include such questions would strengthen the methodological and thematic coherence of the study.

Validity of the findings

The subsection “HRV Through the Decades” provides an insightful and detailed historical narrative on the evolution of HRV research, tracing its roots from the 18th century to its current applications. The chronological structure is effective in highlighting key advancements, and the integration of significant milestones—such as the development of ECGs, Holter monitors, and spectral analysis—adds depth to the discussion. The subsection also successfully connects historical discoveries to their clinical and technological implications, such as the relevance of HRV in fetal distress and the democratization of HRV monitoring through wearable devices. However, a few areas could benefit from further refinement:

Clarity and Balance: While the later decades (1990s onward) are well-detailed, the earlier periods (18th–19th centuries) are less developed. Adding more context to the contributions of pioneers like Stephen Hales and Carl Ludwig would provide a more balanced narrative. For example:
• How did these early findings shape later technological developments?
• What challenges did researchers face in these early studies due to technological limitations?

Integration of Trends: Figure 3 is mentioned but not fully utilized in the text. The subsection would benefit from a more explicit discussion of trends shown in the figure. For example:
• How has the volume of HRV research grown across different decades?
• What shifts in focus (e.g., from gynecology to sports science) are evident, and what factors drove these changes?

Nonlinear Analysis and Recent Advances: The introduction of nonlinear methods, such as Poincaré plots, is mentioned briefly. Expanding on the importance of these methods for modern HRV analysis would enhance the discussion. For example:
• What unique insights do nonlinear methods provide compared to time- or frequency-domain analysis?
• How have these methods impacted clinical or athletic applications?

Literature and Citations: Some claims would benefit from additional references to support them. For example:
• The shift in HRV research focus to cardiovascular health and sports science could be substantiated with recent reviews or studies.
• The statement on HRV’s importance in the 2020s for monitoring recovery, exercise, and stress management should be supported by current literature or market data.

Overall Recommendations:
• Expand the discussion of earlier HRV research milestones to provide a more balanced narrative.
• Integrate Figure 3 more explicitly into the discussion, focusing on trends and shifts in research priorities.
• Elaborate on nonlinear methods and their role in modern HRV analysis.
• Strengthen the subsection with additional references to support key claims.

The subsection "HRV Measurement and Devices" provides a thorough review of the technological advancements and methodologies that have shaped HRV measurement over time, starting with early tools such as galvanometers and kymographs, progressing through Holter monitors, and culminating in modern wearable devices. The discussion highlights key milestones, including the introduction of ECG for precise HRV analysis, the development of portable Holter devices for long-term monitoring, and the growing adoption of PPG-based sensors in wearable technologies. While the narrative effectively outlines these advancements, it could benefit from further integration of historical context, such as the pioneering work of Anastasios Filadelfeus in pulse measurement (Stangl & Riedl, 2022). Additionally, a more detailed comparison of ECG and PPG methods, emphasizing their respective strengths and limitations in clinical and consumer applications, would enhance the analysis. The discussion of recent trends, such as the integration of AI and machine learning for real-time HRV analysis, is promising but underdeveloped. Expanding on how these technologies are transforming HRV research and healthcare applications would add depth. Furthermore, while the section acknowledges the lack of independent clinical validation for many wearable devices, it could delve deeper into the implications of this gap for healthcare adoption. By addressing these points and streamlining repetitive content, the subsection could provide a more balanced, comprehensive, and forward-looking perspective on HRV measurement and devices.

Suggested Reference:
Stangl, F. J., & Riedl, R. (2022). Measurement of heart rate and heart rate variability in NeuroIS research: Review of empirical results. In Davis et al. (Eds.), Information Systems and Neuroscience: NeuroIS Retreat 2022. Springer. https://doi.org/10.1007/978-3-031-13064-9_29

The subsection “HRV Analysis Techniques” provides a thorough overview of HRV analysis techniques, clearly explaining metrics and their physiological relevance. The tables effectively organize these metrics, but it could be enhanced by including columns linking each metric to its physiological source and practical applications, such as cardiovascular risk assessment or stress monitoring. Additionally, it would be beneficial to address that many empirical studies combine metrics from time-domain and frequency-domain analyses to provide a more comprehensive view of autonomic function. Creating a new subsection, such as "time-frequency domain," which lists all identified studies from the review that integrate metrics from both domains, would better reflect current research practices. Moreover, the subsection would benefit from a supplementary table connecting metrics to specific studies, which would strengthen the scientific grounding of the review (this suggestion can be also applied to the following subsection “Confounding Factors”). Practical applications, such as RMSSD’s use in athletic recovery or SDNN’s role in evaluating autonomic function, could also be highlighted to contextualize the metrics’ utility. Clarifying and expanding this would provide a more comprehensive analysis of HRV techniques. These enhancements would ensure the subsection is balanced, practical, and well-supported by literature.

Suggested Reference for Cofounding Factors:
Massaro, S.; Pecchia, L.: Heart Rate Variability (HRV) analysis: A methodology for organizational neuroscience. Organizational Research Methods, Vol. 22 (2019), No. 1, pp. 354–393. https://doi.org/10.1177/1094428116681072

The subsection “HRV in Clinical Medicine” provides a detailed overview of the barriers to integrating HRV into clinical medicine, including the lack of standardization, complexity of data interpretation, and skepticism surrounding wearable devices. While the challenges are well-articulated, the section would benefit from a stronger connection to empirical evidence, such as meta-analyses on HRV’s specificity and sensitivity or studies validating wearable devices. Additionally, the focus on barriers overshadows HRV’s successes, such as its established use in predicting mortality post-myocardial infarction and detecting diabetic autonomic neuropathy. Incorporating a table to summarize HRV’s challenges and potential solutions would improve clarity and accessibility. Furthermore, the section could expand on future directions, such as standardization efforts, AI-driven tools for HRV interpretation, and large-scale clinical trials to validate wearable technology. These enhancements would provide a more balanced and forward-looking perspective on HRV in clinical medicine.

The subsection “Contradictions in the Literature” effectively identifies key contradictions in HRV research, such as the debate over the LF/HF ratio as a measure of sympathovagal balance, the differences between PRV and HRV, and the mathematical relationship between HR and HRV. While these discussions are well-supported by literature, further exploration of their practical implications would enhance the section. For instance, clarifying the conditions under which the LF/HF ratio might still hold value or providing examples of when PRV could be used as a substitute for HRV would add depth. Additionally, a visual aid summarizing these contradictions and their implications would improve clarity. The subsection could also benefit from a more detailed explanation of the hyperbolic and negative exponential models and their relevance to HRV analysis. Finally, discussing future research directions, such as validating PRV against HRV across contexts or refining the LF/HF ratio for specific applications, would make the content more forward-looking. These additions would provide readers with a clearer understanding of the challenges and opportunities in HRV research.

The subsection “Potential of HRV” provides a compelling overview of HRV’s expanding potential in fitness, mHealth, and healthcare, driven by wearable technology and machine learning. It highlights the growing adoption of consumer-grade devices for HRV analysis and their utility in personalized training regimens, as well as the transformative role of machine learning in diagnostics, achieving high accuracy for conditions like obstructive sleep apnea and diabetes. However, the discussion could benefit from a more balanced perspective by addressing challenges such as the variability of HRV norms, the need for standardization in wearable technology, and the integration of mHealth data into clinical workflows. Additionally, exploring intersections between mHealth and AI would add depth, particularly regarding how mobile applications leverage machine learning for real-time HRV analysis. Including a visual summary of HRV applications, benefits, and challenges would enhance clarity and accessibility, making this subsection more comprehensive and forward-looking.

Additional comments

Your manuscript provides a comprehensive and insightful overview of HRV's applications, limitations, and future potential, supported by relevant literature. To further enhance its impact, consider ensuring clarity and consistency by defining abbreviations like HRV, PNS, SNS, and PRV at their first mention in each section. Strengthening integration across sections—for example, linking the contradictions in HRV literature to the challenges of clinical adoption—would provide a more cohesive narrative. Adding visual aids, such as tables connecting HRV metrics to applications or figures summarizing key contradictions, could improve accessibility. Additionally, a concluding paragraph summarizing practical implications and outlining future research directions, such as addressing gaps in wearable device validation or leveraging AI for standardization, would make the manuscript more forward-looking. Simplifying technical language in some areas could also broaden its accessibility to a wider audience. Overall, your work offers a valuable contribution to HRV research, and these refinements would further enhance its clarity and impact.

·

Basic reporting

This manuscript is valuable as a comprehensive overview of heart rate variability (HRV) in time and space. I thoroughly enjoyed reading it.

Experimental design

However, due to the nature of this article as a scoping umbrella review of review articles on HRV, much of the information provided relies on citations from secondary or tertiary sources. Since the intended readers of this paper are likely individuals seeking to apply HRV to their own research or professional practice, it would be more beneficial if specific statements were directly supported by citations from the original studies on which they are based.

Validity of the findings

For example:
1. "In 1981, Akselrod et al. found that analyzing RR interval variability provides physiological insights into cardiac autonomic control (Ishaque et al., 2021)." A reference to Akselrod et al.’s original work is necessary.
2. In 2006, Poincare plots were introduced to visually represent non-linear scatter plots linked to heart conditions and reduced HRV (Ishaque et al., 2021)." To understand Poincare plot analysis in HRV, readers are required to navigate through multiple layers of sources: Ishaque et al. (2021), Trends in Heart-Rate Variability Signal Analysis (Frontiers in Digital Health, 3:639444); Acharya UR et al., Heart Rate Variability: A Review (Medical & Biological Engineering & Computing, 2006, 44:1031–51); and Kamen PW et al., Poincare Plot of Heart Rate Variability Allows Quantitative Display of Parasympathetic Nervous Activity (Clinical Science, 1996, 91:201–208). This approach is overly complex, time-consuming, and lacks clarity.
3. Please review and address similar issues throughout the manuscript.

Additional comments

None.

·

Basic reporting

Line 96 to 105: The precise gaps in HRV research and practice must be clearly stated in the introduction. It references "research gaps" later, but the author doesn't explain what they are or why filling them is so important.

Experimental design

Lines 119 to 122: The manuscript does not include disease-specific reviews, gene/genetic studies, and therapeutic studies. Nonetheless, these fields might offer insightful information about the clinical applicability of HRV. I feel that the authors should specify the reason for exclusion.

Validity of the findings

Line 313: non-linear analysis is mentioned. Please address the gap in the non-linear analysis.
Line 567- 572 Contradictions in the literature, such as the LF/HF ratio controversy, are highlighted throughout the manuscript. Could the writers go into more detail on how these disputes affect academic and healthcare professionals in real-world settings? What potential effects can these arguments have on clinical judgment, data interpretation, or clinical decision-making?"
Line 680 to 682: In the recommendation section, The findings point to gaps in the literature, but they don't provide many recommendations for action. Provide more thorough recommendations for future studies that prioritize specific HRV metrics as recommended by the authors.

Additional comments

nil

---

## Round 0.2 · accepted · Accept

I confirm that the authors have addressed all of the reviewers' comments, improving the quality of the manuscript. Previous reviewers were invited but declined to review the corrections. I checked the corrections required by reviewers.